**Cite this article:** Šćepanović S, Aiello LM, Barrett D, Quercia D. 2022 Epidemic dreams: dreaming about health during the COVID-19 pandemic. *R. Soc. Open Sci.* **9**: 211080. https://doi.org/10.1098/rsos.211080

human–computer interaction

dreams, COVID-19, Twitter, health, medical conditions

**Author for correspondence:**
Sanja Šćepanović
e-mail: sanja.scepanovic@nokia-bell-labs.com

# Epidemic dreams: dreaming about health during the COVID-19 pandemic

Sanja Šćepanović[1], Luca Maria Aiello[2], Deirdre Barrett[3] and Daniele Quercia[1,4]

[1]Nokia Bell Labs, 21 JJ Thomson Avenue, Cambridge CB30FA, UK
[2]Rued Langgaards Vej 7, 2300, Copenhagen, Denmark
[3]Harvard Medical School, 352 Harvard Street, Cambridge, MA 02138, USA
[4]CUSP, King's College London, Strand, London, WC2R 2LS, UK

LMA, 0000-0002-0654-2527

The continuity hypothesis of dreams suggests that the content of dreams is continuous with the dreamer's waking experiences. Given the unprecedented nature of the experiences during COVID-19, we studied the continuity hypothesis in the context of the pandemic. We implemented a deep-learning algorithm that can extract mentions of medical conditions from text and applied it to two datasets collected during the pandemic: 2888 dream reports (dreaming life experiences), and 57 milion tweets (waking life experiences) mentioning the pandemic. The health expressions common to both sets were typical COVID-19 symptoms (e.g. cough, fever and anxiety), suggesting that dreams reflected people's real-world experiences. The health expressions that distinguished the two sets reflected differences in thought processes: expressions in waking life reflected a linear and logical thought process and, as such, described realistic symptoms or related disorders (e.g. *nasal pain, SARS, H1N1*); those in dreaming life reflected a thought process closer to the visual and emotional spheres and, as such, described either conditions unrelated to the virus (e.g. *maggots, deformities, snake bites*), or conditions of surreal nature (e.g. *teeth falling out, body crumbling into sand*). Our results confirm that dream reports represent an understudied yet valuable source of people's health experiences in the real world.

## 1. Introduction

In today's heavily interconnected world, the effects of health crises spread rapidly in the physical space but also in people's minds [1–3], as the latest pandemic of COVID-19 has dramatically demonstrated [4,5]. During its outbreak, various public bodies were able to monitor health concerns and behavioural changes at the scale of entire countries, and did so through extensive surveys [6,7]. However, conventional surveying is costly and has limited

scope and flexibility (even when performed digitally) [8], therefore researchers investigated alternative online sources, especially social media, to infer health information cheaply from public conversations [1,2,9,10]. For example, using social media, researchers have found that the frequency of social media posts expressing anxiety, depression and other mental issues has significantly increased during COVID-19 [11,12].

So far, health studies of COVID-19 based on conversation data have been limited in two main ways. First, they did not deal with comprehensive health concerns; on the contrary, they were tailored to specific health conditions, and many of them relied on simple keyword matching [13], which hinders their generality and adaptability.

Second, the social media data that these studies rely on reflects mostly people's waking-life discussions and experiences, overlooking what people process during their sleep. Dream content has been repeatedly linked to physical and mental well-being [14–17]. That is why researchers have conducted several studies on dreams during the COVID-19 pandemic. However, they either: (i) analysed textual dream reports with sentiment and topical analysis [15,16,18,19] or through standard dream analysis scales [19] or (ii) used surveys to measure the level of psychological distress through questions concerning sleep and dreams, such as quality of rest and nightmare frequency [14,20–24], and did not necessarily entail the collection and analysis of dream reports. So far, no study has conducted a systematic analysis of medical conditions featured in dream reports and compared them with those expressed in waking discussions. The relationship between health concerns discussed during waking life and the representation of such concerns in dreams is relevant to the study of people's well-being because of the similarity between neurobiological mechanisms underlying episodic memory during waking life and those of dream recall during sleep [25]. This is often referred to as 'continuity hypothesis of dreaming' [26]. The hypothesis postulates that dreams are a continuation of our waking experiences, especially of those that trigger vivid emotions. It was first formulated by psychologist Calvin Hall in the 190s, and, in the following decades, it has been extensively studied [26,27].

In this study, we systematically compared textual expressions of health experiences in waking discussions and in dreams.[1] To do so, we resorted to a state-of-the-art deep-learning method [28] capable of extracting mentions of any medical condition from text, and applied it to two main data sources. The first is a collection of 57 milion tweets posted in relation to COVID-19. The second source is a set of written dream reports that describe the dreams of 2888 people during the pandemic. In doing so, we made the following contributions:

— [C1] We defined a set of methodologies that grouped medical conditions into classes based on frequency of occurrences (including those *typical of waking discussions*, those that are *equally present*, and those *typical of dream reports*) and that arranged dream conditions into groups based on semantic similarity (these groups include *fear, anxiety, panic, grief, phobia, trauma, breathing* and *choking* related ones) (§2).
— [C2] In studying similarities between waking discussions and dream reports, we found evidence corroborating the continuity hypothesis of dreaming: indeed, we found that mentions of common COVID-19 symptoms (e.g. *fever, cough, anxiety*) were equally prevalent in both sets (§3.1).
— [C3] We found that waking discussions mentioned concrete medical symptoms more often (e.g. *immunocompromise, influenza, nasal pain*) than what dream reports did. On the other hand, dream reports included more surreal conditions not directly associated with the virus (e.g. *teeth suddenly falling out, deformities* and *body crumbling into sand*), which turned out to be metaphorical embodiments of actual symptoms (§3.2). This apparent deviation from the continuity hypothesis can be explained with the hypothesis that sleep deforms real-life experiences through the lens of emotions and figurative imagery [29].

# 2. Material and methods

To begin with, we discuss our three datasets (§2.1): pre-pandemic tweets, tweets mentioning COVID-19 (waking pandemic discussions), and dream reports collected during the pandemic. Then we discuss our methodology for studying health mentions in these datasets (§2.2).

## 2.1. Datasets collected during and prior to the pandemic

### 2.1.1. Tweets (pre-pandemic and pandemic waking discussions)

As a pre-pandemic baseline, we used an existing dataset of tweets from the period between 1 January and 24 February of 2014 to capture pre-pandemic waking discussions. This dataset consisted of

---

[1]The project webpage is http://social-dynamics.net/dreamscatch/.

974 482 English tweets posted by 240 959 unique users. To then study pandemic waking discussions, we used an existing dataset of 129 911 732 tweets collected through the Twitter Streaming API [30]. The dataset collected by Chen *et al.* includes tweets containing words from a manually curated list of terms related to COVID-19 and is openly shared via GitHub [31]. From this initial dataset, we extracted 57 287 490 English tweets posted in the year 2020 from 1 February to 30 April by 11 318 634 unique users. This timeframe includes the period of the initial spread of the virus until the peak of the number of deaths worldwide during the first wave of infections. The number of active users per day varies from a minimum of 72 000 on 2 February to a maximum of 1.84 million on 18 March, with an average of 437 000. The median number of tweets per user during the whole period is 2. A small number of accounts tweeted a disproportionately high number of times, reaching a maximum of 15 823 tweets; those were automated accounts, which were discarded by our heuristic approach. Given that we selected English tweets, most of the users in our dataset came from the USA and UK. In the USA, 22% of the population use Twitter, and many socio-demographic characteristics of these users deviate from the general population's [32]: compared with the average adult in the USA, Twitter users are wealthier (41% with an income of $75 000 versus 32% in the general population), younger (median age of 40 versus 47), more likely to have college degrees (41% versus 31%), and slightly more likely to identify with the Democratic Party (36% versus 30%). Notably, Twitter users are gender-balanced though (50% women).

### 2.1.2. Dream reports

A survey was posted on 23 March and the responses were downloaded for analysis on 15 July. The survey asked respondents to submit 'any dreams you have had related to the COVID-19 coronavirus'. The survey also enquired about age, gender and nationality. The survey was announced on 11 Facebook groups—three smaller ones (611–7461 members) focused on dreaming, and eight larger ones (27 961–41 351 members) focused on the pandemic. The survey has also been linked from articles in major media in the USA, Europe, South America, Australia, New Zealand and India. The survey is still ongoing at the time of the writing; in this study, we used the responses collected as of 15 July, for a total of 2888 participants. Participants were invited to submit multiple relevant dreams, but only the first dream from each was used, in order not to bias the results towards prolific dream reporters. Respondent's ages ranged from 18 to 91 years, with a median of 40 and a standard deviation of 16.89. Among all the subjects, the vast majority were women (1998), and 68 identified as either gender-neutral or transgender; this subset was deemed not large enough or consistent enough to analyse in the present study, but it may be included in future analyses as the survey N grows. Nationalities, in decreasing frequency, were USA (2011), British (249), Italian (212), Canadian (173), Spanish (91) Indian (54), Peruvian (54), German (46), Mexican (42), Australian (34), Brazilian (33), French (22), Polish (22) and 73 other nationalities with less than 20 respondents each. The dream reports in our analysis are all written in English.

## 2.2. Methods

Our methodology unfolded in three main steps (figure 1). First, on each of the three datasets (i.e. waking pre-pandemic/pandemic discussions and dream reports), we separately applied a deep-learning natural language processing (NLP) method called MedDL [28] to extract mentions of medical conditions (§2.2.1). Second, we classified the extracted conditions based on their relative prevalence in the two pandemic datasets into three groups: conditions typical of waking discussions, those equally prevalent and those typical of dream reports (§2.2.2). Third, we built a co-occurrence network of dream conditions and analysed it using graph modelling to find: important conditions (i.e. conditions linked to many other conditions), and groups of conditions that co-occur frequently (§2.2.3).

### 2.2.1. Extracting medical conditions

To detect health mentions in our textual datasets, we applied a state-of-the-art NLP deep-learning tool called MedDL [28]. The tool is based on deep recurrent neural networks [33] and contextual embeddings [34], and it was developed to extract mentions of any type of medical condition and of phrase stated in the health context from free-form text without the need for pre-processing the text itself. That is to say, on the input of free-form sentences, MedDL directly outputs any medical condition or health reference in those sentences. We trained MedDL on labelled tweets (Micromed

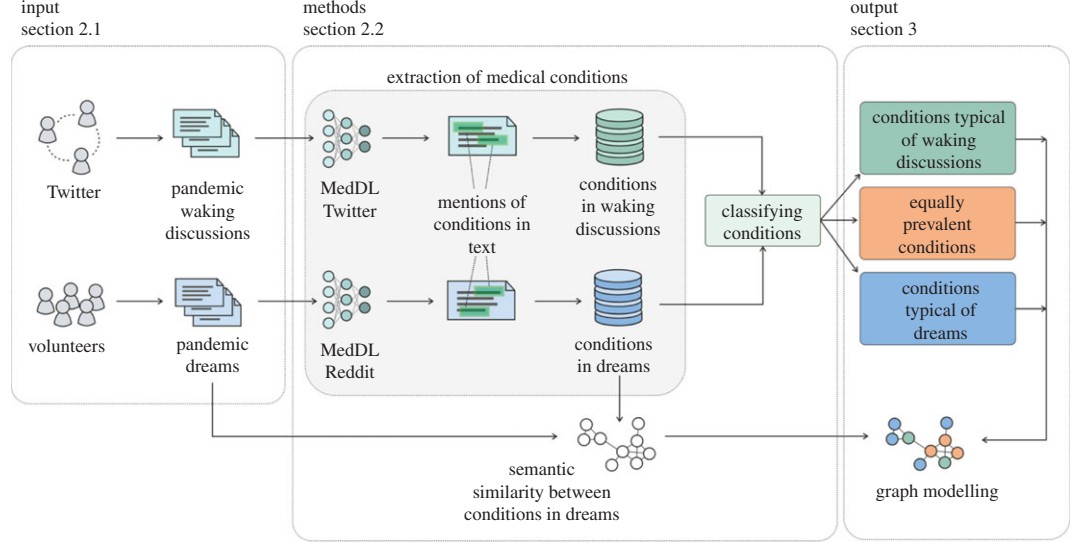

**Figure 1.** Overview of our methodology. On *input*, there are two data sources: waking discussions (tweets), and dream reports. The *methods*: extract medical conditions from input text using MedDL [28] (§2.2.1); classify the extracted conditions based on the relative prevalence in two datasets (§2.2.2); and model the network of condition co-occurrences in dream reports (§2.2.3). In *output*, there are not only three classes of medical conditions but also a semantic similarity graph of conditions in dream reports.

dataset [35]) to parse waking discussions (as they are tweets), and on labelled Reddit posts (MedRed dataset [28]) to parse dream reports (as these reports consist of sentences in a free-form language without any length limitation and, hence, are similar to Reddit posts). Compared with the mean length of 30 words per tweet, the mean length of Reddit posts is 75, and that for dream reports is 120 words. When tested on these standard benchmarks, MedDL outperformed state-of-the-art medical entity extractors by a large margin [28]. When applied to tweets, MedDL achieved an F1-score of 0.74, while the competing approaches ranged from 0.34 to 0.63. When applied to Reddit posts, whose length and structure are more like those of dream reports than those of tweets, MedDL fared an F1-score of 0.71, while the competing approaches hovered between 0.17 and 0.38.

## 2.2.2. Classifying conditions

To discover which medical conditions are more typical of waking discussions as opposed to dream reports, we relied on the frequency of mentions of those conditions in the two datasets. While our two pandemic datasets are comparable in terms of the time period, and in terms of topic (COVID-19), the actual numbers of tweets are several orders of magnitude higher than those of dream reports (table 1). For this reason, in the following analyses, instead of actual counts, we used the ranks of conditions in tweets/ dream reports. First, we ranked conditions by their frequency in each dataset separately. Then we plotted each condition $c_i \in C = \{c_1, \ldots, c_n\}$ with the pair of its ranks $c_i = (r_{\text{waking}}(c_i), r_{\text{dreams}}(c_i))$ as its coordinates. Three main classes of conditions emerged, as shown in figure 2. The first class consist of conditions whose relative frequency is comparable across the two datasets ($r_{\text{waking}}(c_i) \cong r_{\text{dreams}}(c_i)$, lying close to the diagonal); and two classes of conditions that are typically found in one dataset yet only rarely (or never) in the other ($r_{\text{waking}}(c_i) \ll r_{\text{dreams}}(c_i)$ or $r_{\text{waking}}(c_i) \gg r_{\text{dreams}}(c_i)$, lying close to the axes). If a condition was not mentioned in one of the datasets, we assigned it to the class of conditions typical of the other dataset. Note that in figure 2, we could only represent conditions that were found in both datasets, as their rank is not defined if they are not found in one of the datasets.

To *automatically* partition the conditions into these three logical classes, we applied the following procedure. First, we identified the conditions that are present only in one of the two datasets and assigned them to the class of conditions typical of that dataset. Then, on the remaining set of conditions (i.e. those found in both datasets), we applied a linear regression model where $r_{\text{waking}}(c_i)$ is the dependent variable, and we predicted $r_{\text{dreams}}(c_i)$ as the independent variable. Let us denote the predicted value for $r_{\text{waking}}(c_i)$ as $\tilde{r}_{\text{waking}}(c_i)$. For each condition $c_i = (r_{\text{waking}}(c_i), r_{\text{dreams}}(c_i))$, we then calculated its residual $r_i$ as the difference between the 'true value' $r_{\text{dreams}}(c_i)$ and the predicted value $\tilde{r}_{\text{waking}}(c_i)$. This difference equals the distance along $y$-axis from point $c_i$ to the regression line, and is

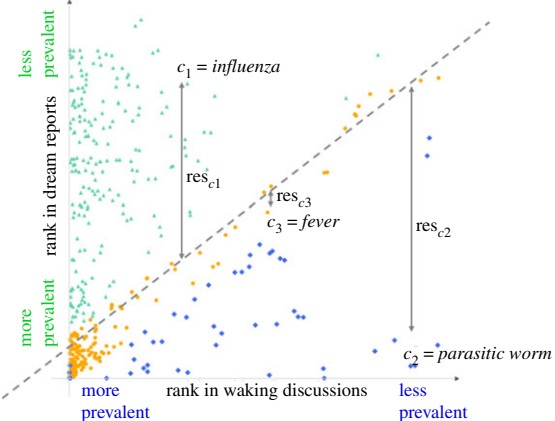

**Figure 2.** Our methodology for classifying conditions based on relative prevalence in two datasets. Each dot represents a condition. On *x*-axis, the condition's rank in waking discussions and on *y*-axis its rank in dream reports is shown. The *higher the rank*, the *less prevalent* the condition in the dataset. The dashed grey line shows the final regression line after the application of the iterative procedure described in §2.2.3. For three selected conditions $c_1$, $c_2$ and $c_3$, one from each of the classes, their residual values from the regression line $res_{c1}$, $res_{c2}$, and $res_{c3}$ are visualized using the grey lines. For $c_1 = $ *influenza*, its residual is positive ($r_{c1} > 0$) because its rank in dream reports is considerably higher than in waking discussions. On the other hand, for $c_2 = $ *parasitic worm*, its residual is negative ($r_{c2} < 0$) because its rank in the dream reports is considerably lower than in waking discussions. Finally, for $c_3 = $ *fever*, its residual is close to zero ($r_{c3} \sim 0$) because its two ranks in the two datasets are similar ($r_{waking}(c_3) \cong r_{dreams}(c_3)$). Upon application of the classification method, all the conditions get classified into the three shown classes; green: conditions *typical of waking discussions*; orange: *equally prevalent* and blue: conditions *typical of dream reports*.

**Table 1.** Statistics of the three datasets.

|  | pre-pandemic waking discussion | pandemic waking discussion | dream reports |
|---|---|---|---|
| time period | 1 Jan–24 Feb 2014 | 1 Feb–30 Apr 2020 | 23 Mar–15 July 2020 |
| no. tweets/dream reports | 974 482 | 57 287 490 | 2888 |
| no. users | 240 959 | 11 318 634 | 2888 |
| no. conditions | 12 177 | 2 606 500 | 3748 |
| no. unique conditions | 2202 | 24 248 | 1732 |

positive for points above the regression line, negative for points below it, and close to zero for the points along the line. To then separate the points into three classes, we calculated the mean residual value across all the points $c_i$, denoted as $\mu$, and the corresponding standard deviation, denoted as $\sigma$. We then assigned the conditions with a too large positive residual (res $- \mu \geq 1.5 * \sigma$) to the class of conditions *typical of waking discussions* because for them $r_{waking}(c_i) \ll r_{dreams}(c_i)$, and the conditions with a too large negative residual ($\mu - res \leq -1.5 * \sigma$) to the class of conditions *typical of dream reports* because for them $r_{dreams}(c_i) \ll r_{waking}(c_i)$. The conditions for which the residual was close to zero ($-1.5 * \sigma \leq res - \mu \leq 1.5 * \sigma$) were assigned to the *equally prevalent* class, because for them $r_{waking}(c_i) \cong r_{dreams}(c_i)$. This procedure resulted in separating the most extreme points from the ones closer to the diagonal. To obtain the final separation into the three groups shown in figure 2, we needed to recursively repeat the procedure by fitting a new linear regression model to the points in the *equally prevalent class*, until all the remaining conditions in that class were fit to an almost straight line. Specifically, we started by considering all the points and fitting the first regression line, which enabled us to remove the points that had the highest residuals. We then fitted a new linear regression line on the remaining points, which would be closer to the diagonal line, and again removed the outliers. The procedure finished when all the remaining points remained on an almost straight line (figure 2). This method found 21 551 conditions in waking discussions only, 1335 in dream reports only, 277 more prevalent in waking discussions, 57 more prevalent in dream reports and 63 equally prevalent.

## 2.2.3. Finding important conditions and groups of conditions based on semantic similarity

As we shall discuss in the results, conditions that are more prevalent in dream reports are often characterized by unreal imagery. To discover how those special types of conditions are related to concrete medical conditions, we constructed a co-occurrence graph of the 1732 unique medical conditions in dream reports. In this graph, nodes represent all the medical conditions extracted from the dream reports. Two nodes are connected by an edge if they both appeared in the same dream report. Edges are weighted by the number of dreams in which the two conditions co-occurred. We used this co-occurrence network for two main purposes. First, to capture the semantic relatedness of medical conditions: symptoms or diseases that were often mentioned together are likely to describe the same condition, and second, to discover groups of related conditions as, in the network, they form densely connected clusters of nodes. Out of the 1732 unique medical conditions in dream reports, 313 did not co-occur with any other condition (i.e. they resulted in singleton nodes in the graph), so we discarded them. The graph containing the remaining 1419 nodes had 4084 edges.

To assess the robustness of the association between pairs of linked conditions and filter out pairs whose frequency of co-occurrence was indistinguishable from the outcome of a random process, we calculated two measures that are commonly used in the study of comorbidity networks: relative risk $RR_{ij}$, and the Phi correlation coefficient $\varphi_{ij}$ [36]. These are

$$RR_{ij} = \frac{P_{ij} \cdot N}{P_i P_j - P_{ij}}$$

and

$$\phi_{ij} = \frac{P_{ij} N - P_i P_j}{\sqrt{(P_i P_j - C_{ij})(N - P_i)(N - P_j)}} \, ,$$

where $N$ is the total number of dream reports, $P_i$ (or $P_j$) is the number of these reports mentioning condition $i$ (or $j$) and $P_{ij}$ is the number of those mentioning both conditions $i$ and $j$. When two conditions co-occur more frequently than expected by chance, $RR_{ij} > 1$ and $\varphi_{ij} > 0$. The statistical significance of $\varphi_{ij}$ can be determined through a $t$-test resulting in a $p$-value ($p_\varphi$)

$$t = \frac{\phi_{ij}\sqrt{N-2}}{\sqrt{1 - \phi_{ij}^2}}.$$

The significance of $RR_{ij}$ can be assessed by estimating its 99% confidence intervals whose upper bound (UB) and lower bounds (LB) are

$$UB(RR_{ij}) = RR_{ij} \cdot e^{(2.56 \cdot \sigma_{ij})},$$
$$LB(RR_{ij}) = RR_{ij} \cdot e^{(-2.56 \cdot \sigma_{ij})}$$

and

$$\sigma_{ij} = \frac{1}{P_{ij}} + \frac{1}{P_i P_j} - \frac{1}{N} - \frac{1}{N^2}.$$

The two measures are not independent of each other yet tend to be complementary. We used both to filter out network links that correspond to non-robust associations. Specifically, we preserved only the links that satisfied two conditions

$$\phi_{ij} > 0 \; \wedge \; p_\phi < 0.01$$

and

$$LB(RR_{ij}) > 1.$$

This filtering step reduced the network from 1419 nodes linked by 4084 edges to 1416 nodes linked by 3759 edges.

To assess the relative position of the three classes of conditions that we previously identified (§2.2.3) in the network, we measured two quantities. First, we calculated the centrality of each node in the graph—a measure of how 'important' and well-connected it is. Specifically, we used PageRank centrality, a widely used centrality measure that estimates the likelihood that a 'walker' who travels from node to node by picking random edges will end up in the specific node. The idea is that a condition with a high (low) PageRank is more (less) important because it is more (less) well-connected

to others. Second, we computed the attribute assortativity of the network with respect to the three node classes. Assortativity measures the propensity of network's nodes belonging to a given class to be linked to other nodes of the same class. Assortativity scores can be positive or negative, which indicate assortative and disassortative networks, respectively. The idea is that in a network with high (low) assortativity the nodes tend to be linked with other nodes from the same (different) class. In our case, this can inform whether the nodes from the three different classes tend to appear in the same or different dream reports.

# 3. Results

As we will shortly see, we found that common COVID-19 symptoms were present in both waking discussions during the pandemic (and not in those pre-pandemic) and dream reports (§3.1). Yet, we also found that sleep deformed real-life experiences, in that dream reports tended to contain metaphorical embodiments of actual symptoms (§3.2).

## 3.1. Common COVID-19 symptoms present in both waking discussions and dream reports

MedDL found 12 177 health conditions in pre-pandemic waking discussions (unique 2202), and 2 606 500 conditions in pandemic waking discussions (24 248 unique) and 3748 conditions in dream reports (1732 unique). In all cases, we found heavy-tailed frequency distributions: a small number of frequent conditions was mentioned tens of thousands of times in the case of tweets, and hundreds of times in the case of dream reports (table 2), while the majority of conditions were mentioned only once or a few times.

We first compared conditions in pandemic waking discussions to the pre-pandemic ones. While the top frequent conditions pre-pandemic included *tiredness*, *pain* and *hangover*, they were expectedly overtaken by the mentions of COVID-19 symptoms during the pandemic. From the top pre-pandemic conditions, those that stayed among the top frequent during the pandemic are *stress*, *anxiety* and *cancer*.

Upon classifying pandemic conditions in waking discussions versus dream reports (§2.2.2), we found that those commonly mentioned in both datasets included: *coronavirus*, *anxiety*, *cancer*, *coughing* and *stress*. Conditions such as *infectious disease* and *ebola* were highly ranked among the top mentions in waking discussions, but not in dreams, whereas mentions of *seasick* and *bleeding* were more frequent in dream reports. Some of the rare conditions found in tweets included *hypergammaglobulinemia*, *hyperreactivity pulmonary destruction* or *vaping-related lung illnesses*, while rare conditions in dreams included *feels like water fills my lungs*, *extreme déjà vu*, or *grabbing at my throat and swollen tongue*. In summary, in the group of equally prevalent conditions in waking discussions and dream reports, we found different mentions of COVID-19 itself (e.g. *coronavirus*, *corona virus*, *covid* or *virus*), or of its common symptoms (e.g. *fever*, *pain*, *sore throat*, *migraine* and *cough*) [37].

## 3.2. Deformation of real-life experiences during sleep

In addition to the conditions common to the two sets, there were conditions that differentiated the two. We found that symptoms found more in waking discussions were realistic symptoms directly linked to COVID-19 (e.g. *body aches*, *abnormal heart rate*, *bronchitis*, *pneumonia* and *nasal pain*), or related to similar conditions (e.g. *influenza*, *SARS*, *H1N1*, *bird flu*, *allergy*, *flu like symptoms*), showing that people were discussing other infectious diseases as well. On the other hand, conditions mentioned mostly in *dream reports* included those that did not actually occur with the virus (*maggots*, *deformities*, *red virus* and *snake bites*) or surreal ones (*teeth suddenly falling out*, *body crumbling into sand* and *rodents moving around in my periphery*), probably reflecting an exaggerated visual depiction of something wrong with the body or of dramatic scenarios resulting from the virus (table 3).

To explore this further, we built the co-occurrence graph of the conditions expressed in dreams (§2.2.3). After filtering, the network contained 1416 nodes and 3759 edges. Out of all the nodes, 60 were conditions equally prevalent in waking discussions and dream reports, 232 were more typical of waking discussions than dreams, 49 were more typical of dreams than waking discussions, and 1075 were found only in dream reports. The giant connected component of the graph (a sub-graph in which there exists a path that connects each node to every other node) contained 1139 nodes and 3447 edges; the remaining nodes were scattered in other 104 small, isolated components. The average PageRank centrality was highest for the medical conditions common to waking discussions and dream reports (average $pr = 111 \times 10^{-3}$), lowest for waking discussions-specific conditions (average

**Table 2.** Top frequent conditions in the three datasets.

| pre-pandemic waking discussions | no. tweets | pandemic waking discussions | no. tweets | dream reports | no. dream reports |
|---|---|---|---|---|---|
| tired | 972 | coronavirus | 1 778 456 | virus | 386 |
| pain | 659 | flu | 67 712 | coronavirus | 242 |
| cancer | 536 | corona virus | 21 576 | COVID | 187 |
| hungry | 418 | sick | 17 745 | anxiety | 128 |
| stress | 400 | bronchitis | 11 461 | COVID 19 | 100 |
| hangover | 340 | infectious disease | 10 441 | nightmares | 64 |
| sick | 340 | swine flu | 9291 | pandemic | 58 |
| cold | 299 | fever | 9158 | coughing | 58 |
| headache | 235 | influenza | 8144 | corona virus | 41 |
| sore | 198 | COVID19 | 7647 | fever | 40 |
| hungover | 185 | cancer | 7133 | corona | 35 |
| ill | 136 | viruses | 7066 | stress | 34 |
| disabled | 132 | ebola | 6941 | cough | 25 |
| depression | 127 | cough | 5820 | pain | 24 |
| flu | 99 | anxiety | 5223 | COVID19 | 21 |
| exhausted | 88 | coughing | 5060 | trouble breathing | 17 |
| anxiety | 86 | pneumonia | 5034 | infection | 17 |
| migraine | 85 | HIV | 4899 | bleeding | 15 |
| heart attack | 85 | virus | 4868 | panic attack | 15 |
| obesity | 72 | stress | 4768 | asthma | 14 |
| insomnia | 67 | allergy | 4657 | cancer | 13 |

pr $= 9 \times 10^{-3}$) and in-between for dream-specific conditions (average pr $= 20 \times 10^{-3}$). The average PageRank of the class of equally prevalent conditions is one or two orders of magnitude higher than that of the other two classes. This indicates a strong core-periphery structure where the conditions equally prevalent in waking discussions and dream reports act as hubs that connect groups of dataset-specific conditions. The network is slightly disassortative (assortativity = −0.12), meaning that a node belonging to a given class is more likely to connect to nodes of a different class than what would happen by random chance. Taken together, these results reveal that the 'backbone' of central nodes in the network contains mostly conditions that are equally present in the two datasets, and those central nodes are linked with conditions from different classes (i.e. typical either of dream reports or of waking discussions) (figure 3).

One can observe such structural patterns also by visually inspecting the network. Figure 3 shows the 'core' of the co-occurrence graph, represented by the set of most connected nodes (those with five edges or more). The central nodes represent few clearly identifiable symptoms that are commonly discussed in the context of COVID-19; these include physical conditions (*cough, fever, shortness of breath*) as well as psychological ones (*depression, fear, nightmare*). On the one hand, we have terms scattered around the network. These mostly come from waking discussions and reflect formal medical conditions (e.g. *pneumonia, influenza, bronchitis*).

On the other hand, we have groups of terms directly attached to the central nodes. These represent health conditions predominantly found in dream reports. They can be grouped into eight main groups related to: *choking, breathing, fear, anxiety, panic, grief, trauma* and *phobias*. Symptoms in these groups often add specificity and nuances to the backbone nodes they are connected to. For instance, *anxiety* is associated with *heart racing* and *clenched jaws*, and *breathing* with sleep disorders such as *sleep paralysis*. Other groups include unconventional associations. For example, mentions of *panic attacks* co-occur with descriptions of *numb lips*, while *choking* co-occur with figurative descriptions of suffocation (e.g. *feeling close to the edge*). A group linked to *depression* consists of terms related to *grief* (e.g. *anguish*,

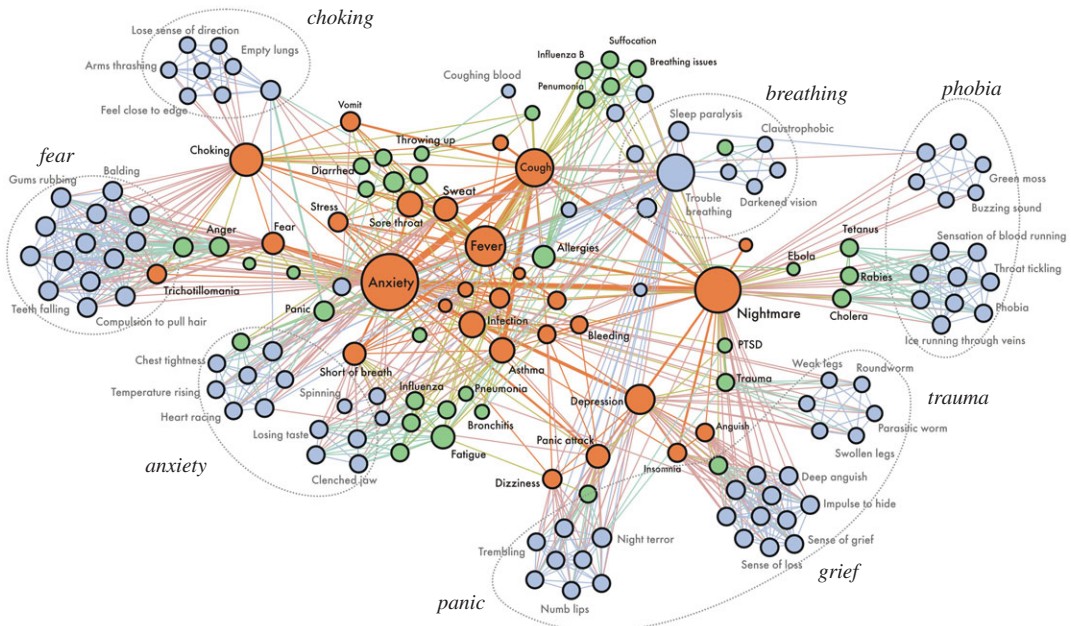

**Figure 3.** The core of the medical conditions co-occurrence network of dream reports. Nodes are sized proportionally to their PageRank centrality in the network and coloured according to the class they belong to (orange: *equally prevalent* in waking discussions and dream reports; green: *typical of waking discussions*; blue: *typical of dream reports*). Edges are scaled proportionally to their weight, which encodes the number of times two conditions co-occurred in the same reports. The names of selected conditions are reported (and these in light grey are the conditions that are more prevalent in dream reports). A few clusters of semantically similar conditions are marked with dashed circles.

**Table 3.** Conditions in the three classes of medical conditions resulting after classification based on the relative prevalence in pandemic waking discussion versus dream reports (§2.2.3). Conditions in each class are ranked from the most frequent (top) to the least frequent (bottom).

| equally prevalent | typical of waking discussions | typical of dream reports |
|---|---|---|
| corona virus | infectious disease | sleep paralysis |
| sick | influenza | trouble breathing |
| cough | HIV | coughing up blood |
| virus | AIDS | gasping for air |
| bronchitis | common cold | spitting out teeth |
| fever | heart disease | thickness or pressure in my chest |
| COVID19 | bird flu | maggot |
| cancer | mental illness | seizure disorder |
| allergy | lupus | social anxiety |
| cold | immunocompromise | overdose |
| stress | measles | overwhelmingly large eyes |
| infection | malaria | teeth started falling out |
| anxiety | mental health | heart was beating out of my chest |
| pneumonia | Lyme disease | teeth breaking off and coming out |
| plague | nasal pain | alien invasion |

*sense of loss*), a concept that did not appear in our waking discussions. Last, some groups contain surreal conditions, which were physical embodiments of mental health conditions known to dream analysts [38]: conditions related to *phobia* included *green moss* and *buzzing sounds*; *trauma* included *worms* and *swollen legs*; and *fear* included grotesque conditions (e.g. *teeth* and *hair* suddenly falling out).

# 4. Discussion

Some of our results match findings previously reported by survey-based studies on the effect of the pandemic on sleeping or dreaming, speaking to the external validity of our results. At the beginning of the pandemic, a study asked 1091 participants in Italy to not only score their dream frequency but also characterize their dreams in terms of emotions, vividness, bizarreness and length [14]. Like our results, compared with pre-lockdown period, increased emotional load and higher frequency of nightmares were reported. Increased anxiety was reported too, in agreement with the four themes found in our network analysis: panic, anxiety, phobia and fear. Another survey with 90 adults asked participants to categorize the content of their dreams according to an adaptation of the Typical Dream Questionnaire [22].

The finding that waking discussions and dream reports contain proportionally equal mentions of COVID-19 itself and of its major symptoms is consistent with the continuity hypothesis of dreaming [26]. It indicates that the waking and dreaming minds are equally focused on the threat of the COVID-19 pandemic—it is the resulting associations and style of thinking about it that are so divergent. The differences in frequency for other categories support the idea that the mind is thinking about this concern in two very different states consistent with what we know about brain activation in waking versus sleep states [39]. That waking discussions contain more references to similar diseases by name and to realistic symptoms of other disorders reflects what people use in referencing known facts to try to figure out more about the threat posed by COVID-19 via a linear, logical process. The phrases more frequent in the dream reports about bizarre body dysfunctions represent a metaphoric manner of thinking about COVID-19. This is consistent with the observation that dreams are generally concerned with the same topics but filtered through distinctive brain states during sleep [40]. The network of symptom co-occurrence sheds light on how the waking and dreaming associations to aspects of COVID-19 diverge. Anxiety and choking link to each other and have equally dense links to other symptoms for both waking and dreaming data. However, for the waking discussions, *anxiety* and *choking* also link to other potential realistic symptoms such as *diarrhoea* and *throwing up*. The dreams' *anxiety* and *choking* references link to symptoms such as *balding* and *teeth falling out*. Interestingly, the symptom *nightmare* is equally densely linked in both waking and dream networks. In the waking discussions, however, it links to potential causes of nightmares: *trauma* and *PTSD*. For the dream reports, it links to the imagery of nightmares: *dark clouds, a buzzing sound, and ice running through veins*. This is a further indication of the waking rational and verbal versus the dreams' visual, emotional and metaphoric approach to dealing with the same issue.

# 5. Conclusion

This study has both theoretical and practical implications. From a theoretical standpoint, we found evidence corroborating two hypotheses put forward in dream science research: the 'continuity' hypothesis of dreaming stating that dreaming life is a continuation of waking life [26], and the 'sleep-to-remember' hypothesis stating that sleep is necessary or at least advantageous to consolidate emotional memories [41]. From a practical point of view, by applying a state-of-the-art algorithm for medical entity extraction to dream reports, we showed that it is possible to integrate the content of dream reports into frameworks of digital health surveillance, which traditionally focused only on aspects related to waking life [42]. Specifically, uncovering uncommon and unrealistic health conditions appearing in dreams can help to track people's hidden worries, emotions and psychological symptoms that are not expressed in their waking lives but are affecting them.

This study has three main limitations that call for future work. The first comes from the potential biases introduced by the data collection process. We collected dream reports specifically concerning COVID-19 (i.e. the users were primed to talk about it), while tweets were written by users who freely decided to vent about COVID-19 (i.e. they self-selected for it). That is why we used a rank-based method to compare the health mentions in the two datasets.

The second limitation has to do with our method for extracting mentions of medical conditions from text. Although MedDL is a state-of-the-art tool with top-class accuracy, its output is not error-free. Because MedDL was trained on social media data only, misclassifications could be more frequent when applied to the dream reports dataset. Our qualitative analysis did not produce evidence for any systematic error that would compromise our results. However, future work could collect additional training data specific to dream reports.

The third limitation has to do with the quality and scope of our two datasets. Our Twitter dataset, albeit large, is not fully representative of the general population. Studies on Twitter are exposed to issues of data noise [43,44], representativeness [45] and self-presentation biases [46]. In the USA, the country in which Twitter has highest penetration rate, socio-demographic characteristics deviate from the general population: Twitter users are much younger, with a higher level of formal education, and are more likely to support the Democratic Party [32]. Our collection of dream reports has limitations too. We gathered the reports through a web survey without imposing limits or constraints on the input text, which could introduce some noise in the data. In some reports, for example, participants reported their feelings about their dreams rather than the dreams themselves. Furthermore, the dream reports were written in English, but they came from participants in different nations. In our study, we treated all the reports as a homogeneous set, thus not accounting for cultural differences or for differences in experiences of the very same pandemic. Currently, the number of dream reports is not large enough to allow for a breakdown by country, but future data collections could be targeted to specific geographical regions and allow just for that.

Ethics. The dream reports were not linked to any personal identifiable information. The participants were informed that the dream descriptions they contributed could have been used for research and analysed with automated tools.

Data accessibility. The text of the dream reports cannot be shared as that would violate the agreement with the study participants. The list of medical conditions across the five categories along with their frequencies, and the network of medical conditions are available on Dryad: https://doi.org/10.5061/dryad.r7sqv9scc, and on the project website: http://social-dynamics.net/dreamscatch.

Authors' contributions. S.S.: conceptualization, data curation, formal analysis, investigation, methodology, resources, software, validation, visualization, writing—original draft, writing—review and editing; L.M.A.: conceptualization, data curation, formal analysis, investigation, methodology, resources, software, supervision, validation, visualization, writing—original draft, writing—review and editing; D.B.: conceptualization, investigation, resources, supervision, validation, writing—original draft, writing—review and editing; D.Q.: conceptualization, formal analysis, investigation, methodology, resources, supervision, validation, writing—original draft, writing—review and editing.

All authors gave final approval for publication and agreed to be held accountable for the work performed therein.

Competing interests. We declare we have no competing interests.

Funding. Authors did not receive any funding other than their own salary.

Acknowledgments. The authors thank to Ke Zhou for helpful discussions.

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
