## [Peer Review File · Royal Society Open Science]

Review History

RSOS-211080.R0 (Original submission)

Review form: Reviewer 1

Is the manuscript scientifically sound in its present form?

No

Are the interpretations and conclusions justified by the results?

No

Is the language acceptable?

Yes

Do you have any ethical concerns with this paper?

No

Have you any concerns about statistical analyses in this paper?

No

Recommendation?

Major revision is needed (please make suggestions in comments)

Comments to the Author(s)

This descriptive study investigated by means of a state-of-the-art deep-learning algorithm mentions of medical condition from text and applied it to two sets of data collected during the COVID-19 pandemic. The authors considered 2,888 dream reports and 57M tweets mentioning the pandemic (waking life experiences).

Not surprisingly the study reports that the health expressions (e.g., coronavirus, anxiety, coughing, and stress), were shared by both sets of data and suggested that dreams reflected people's real-world experiences. Quite obviously, they also found differences in thought processes within the two sets of data.

The results have been interpreted within the context of the the continuity hypothesis of dreams.

Within the intrinsic limits of a scarce originality and of some intrinsic bias in the used methods, the study provides a qualitative picture of dreams during the pandemic.

Major points

1. I understand that the authors are not sleep researchers but is simply unacceptable the equivalence between REM sleep and dreaming. The authors have to realize that empirical research on dreams has definitely clarified that dreams are not exclusive of REM sleep and that they can be collected practically in any stage of sleep. This means that all points in the manuscript in which they speak on REM sleep as a physiological scenario of dream production should be changed accordingly

2. Since a large body of cross-sectional and longitudinal studies on the pandemic has been published using surveys, these findings on dream activity should be discussed compared to the current descriptive data

Scarpelli et al. The impact of the end of COVID confinement on pandemic dreams, as assessed by a weekly sleep diary: a longitudinal investigation in Italy. *Journal of Sleep Research*. - (2021). <https://doi.org/10.1111/jsr.13429>

Gorgoni et al. Pandemic dreams: quantitative and qualitative features of the oneiric activity during the lockdown due to COVID-19 in Italy. *Sleep Medicine*. 2021,81: 20-32. <https://doi.org/10.1016/j.sleep.2021.02.006>.

Scarpelli et al. Pandemic nightmares: Effects on dream activity of the COVID-19 lockdown in Italy. *J Sleep Res*. 2021; 30, (5) e13300 <https://doi.org/10.1111/jsr.13300> I.F.= 3.623

Conte et al. Changes in dream features across the first and second waves of the Covid-19 pandemic. *J Sleep Res*. 2021 Jun 22:e13425. doi: 10.1111/jsr.13425.

Scarpelli et al. Dream activity in narcoleptic patients during the COVID-19 lockdown in Italy. *Frontiers in Psychology*. 2021, 12, 1907 DOI=10.3389/fpsyg.2021.681569

3. Concerning the "continuity hypothesis", it commonly refers to: (1) dreams are a continuation of our waking experiences; (2) neurobiological mechanisms underlying episodic memory in wakefulness are mostly similar to those of dream recall during sleep (e.g., Scarpelli et al. Investigation on Neurobiological Mechanisms of Dreaming in the New Decade. *Brain Sci*. 2021 Feb 11;11(2):220. doi: 10.3390/brainsci11020220). The article only mention the first meaning of this accepted meaning

4. Please, substantiate the sentence in the discussion (page 10, rows 21.22): " The phrases more frequent in the dream reports about bizarre body disfunctions represent a metaphoric manner of

thinking about COVID-19 with more activation in visual and emotional areas and less in verbal and logical ones."

The issue of greater activation in visual and emotional areas is still debated. Please, provide references. Still more, I think that there is no empirical study showing that there is smaller activation "in verbal and logical ones."

5. The dramatic increase of nightmares during the pandemic is one of the most consistent findings of empirical studies and have a relevant clinical implication in relation to PTSD symptoms. The current qualitative findings on nightmares should also be discussed with reference to this evidence.

Review form: Reviewer 2

Is the manuscript scientifically sound in its present form?

Yes

Are the interpretations and conclusions justified by the results?

Yes

Is the language acceptable?

Yes

Do you have any ethical concerns with this paper?

No

Have you any concerns about statistical analyses in this paper?

No

Recommendation?

Accept with minor revision (please list in comments)

Comments to the Author(s)

In this work, the authors present a novel work on dream reports, highlighting their importance, if interpreted correctly, to better understand and get insights about people's health experiences in the real world, specifically in the case of extremely challenging and distressing events, such as the COVID-19 pandemic. Data about dream reports regarding dreaming life experience, and tweets mentioning the pandemic, i.e., waking life experiences, have been collected and used in their study. The target has been that of extract mentions of virtually any medical condition from text by applying deep learning algorithms. Their findings demonstrate how dreams reflected people's real-world experiences and that the health expressions that distinguished the two sets of data reflected differences in thought processes.

The idea of studying the dream reports along with social media posts and analyzing their impact on large-scale phenomena, such as the COVID-19 Pandemic, is extremely interesting. Moreover, this is the first study carrying out a systematic analysis of medical conditions featured in dream reports and compared them to those expressed in waking discussions, starting from the well-known "continuity hypothesis of dreaming". The latter refers to the relationship between health concerns discussed during waking life and the representation of such concerns in dreams, which is relevant to the study of people's well-being.

As the same authors underline, one of the main limitations is in terms of biases linked with the data collection process, but they used a rank-based method to compare the health mentions in the two datasets, by ranking conditions by their frequency in each dataset separately.

In terms of methodology, they used MedDL for extracting mentions of medical conditions from text, and they trained the method on social media data, and as the same authors underline, training on dream reports could be beneficial for the analysis results.

Although the methodology is not novel in terms of NLP deep-learning tool used for extracting mentions of medical conditions, and the data collected and training process may produce biases in results, both the work and findings are novel.

The main aspect on which the authors could focus is on comparing the tool used with other methods to better validate the model. Moreover, a figure showing the pipeline of the methodology would help in the readability.

Finally, to strengthen the methodological approach and extending results, starting from the co-occurrence network of dream reports, which resembles the comorbidity networks (see the work: Moni, Mohammad Ali, and Pietro Liò. "Network-based analysis of comorbidities risk during an infection: SARS and HIV case studies." *BMC bioinformatics* 15.1 (2014): 1-23.), authors could explore some measures, such as the relative risk and ϕ -correlation. Respectively, these measures represent the relationship between the two conditions related to the datasets and the robustness of this association.

Decision letter (RSOS-211080.R0)

Dear Dr Aiello

The Editors assigned to your paper RSOS-211080 "Epidemic Dreams: Dreaming about health during the COVID-19 pandemic" have now received comments from reviewers and would like you to revise the paper in accordance with the reviewer comments and any comments from the Editors. Please note this decision does not guarantee eventual acceptance.

Please submit your revised manuscript and required files (see below) no later than 21 days from today's (ie 10-Nov-2021) date. Note: the ScholarOne system will 'lock' if submission of the revision is attempted 21 or more days after the deadline. If you do not think you will be able to meet this deadline please contact the editorial office immediately.

on behalf of Professor Yulan He (Associate Editor) and Marta Kwiatkowska (Subject Editor)
 openscience@royalsociety.org

Associate Editor Comments to Author (Professor Yulan He):

Comments to the Author:

While the reviewers found that the idea of studying the dream reports along with social media posts to conduct qualitative analysis of dreams during the pandemic is interesting, they raised a few major concerns that should be addressed in the revised manuscript. Examples include more accurate use of terminologies (such as REM sleep, continuity hypothesis), discussions of findings on dream activities in comparison to the existing descriptive data, more empirical studies to support some of the statements in the manuscript, additional measures to quantify the associations revealed by the co-occurrence network of dream reports, etc. Please refer to the reviews for the details of the major points raised

Reviewer comments to Author:

Reviewer: 1

Comments to the Author(s)

This descriptive study investigated by means of a state-of-the-art deep-learning algorithm mentions of medical condition from text and applied it to two sets of data collected during the COVID-19 pandemic. The authors considered 2,888 dream reports and 57M tweets mentioning the pandemic (waking life experiences).

Not surprisingly the study reports that the health expressions (e.g., coronavirus, anxiety, coughing, and stress), were shared by both sets of data and suggested that dreams reflected people's real-world experiences. Quite obviously, they also found differences in thought processes within the two sets of data.

The results have been interpreted within the context of the the continuity hypothesis of dreams.

Within the intrinsic limits of a scarce originality and of some intrinsic bias in the used methods, the study provides a qualitative picture of dreams during the pandemic.

Major points

1. I understand that the authors are not sleep researchers but is simply unacceptable the equivalence between REM sleep and dreaming. The authors have to realize that empirical research on dreams has definitely clarified that dreams are not exclusive of REM sleep and that they can be collected practically in any stage of sleep. This means that all points in the manuscript in which they speak on REM sleep as a physiological scenario of dream production should be changed accordingly
2. Since a large body of cross-sectional and longitudinal studies on the pandemic has been published using surveys, these findings on dream activity should be discussed compared to the current descriptive data

Scarpelli et al. The impact of the end of COVID confinement on pandemic dreams, as assessed by a weekly sleep diary: a longitudinal investigation in Italy. *Journal of Sleep Research*. - (2021). <https://doi.org/10.1111/jsr.13429>

Gorgoni et al. Pandemic dreams: quantitative and qualitative features of the oneiric activity during the lockdown due to COVID-19 in Italy. *Sleep Medicine*. 2021,81: 20-32. <https://doi.org/10.1016/j.sleep.2021.02.006>.

Scarpelli et al. Pandemic nightmares: Effects on dream activity of the COVID-19 lockdown in Italy. *J Sleep Res*. 2021; 30, (5) e13300 <https://doi.org/10.1111/jsr.13300> I.F.= 3.623

Conte et al. Changes in dream features across the first and second waves of the Covid-19 pandemic. *J Sleep Res*. 2021 Jun 22:e13425. doi: 10.1111/jsr.13425.

Scarpelli et al. Dream activity in narcoleptic patients during the COVID-19 lockdown in Italy. *Frontiers in Psychology*. 2021, 12, 1907 DOI=10.3389/fpsyg.2021.681569

3. Concerning the "continuity hypothesis", it commonly refers to: (1) dreams are a continuation of our waking experiences; (2) neurobiological mechanisms underlying episodic memory in wakefulness are mostly similar to those of dream recall during sleep (e.g., Scarpelli et al. Investigation on Neurobiological Mechanisms of Dreaming in the New Decade. *Brain Sci*. 2021 Feb 11;11(2):220. doi: 10.3390/brainsci11020220). The article only mention the first meaning of this accepted meaning

4. Please, substantiate the sentence in the discussion (page 10, rows 21.22): " The phrases more frequent in the dream reports about bizarre body disfunctions represent a metaphoric manner of thinking about COVID-19 with more activation in visual and emotional areas and less in verbal and logical ones."

The issue of greater activation in visual and emotional areas is still debated. Please, provide references. Still more, I think that there is no empirical study showing that there is smaller activation "in verbal and logical ones."

5. The dramatic increase of nightmares during the pandemic is one of the most consistent findings of empirical studies and have a relevant clinical implication in relation to PTSD symptoms. The current qualitative findings on nightmares should also be discussed with reference to this evidence.

Reviewer: 2

Comments to the Author(s)

In this work, the authors present a novel work on dream reports, highlighting their importance, if interpreted correctly, to better understand and get insights about people's health experiences in the real world, specifically in the case of extremely challenging and distressing events, such as the COVID-19 pandemic. Data about dream reports regarding dreaming life experience, and tweets mentioning the pandemic, i.e., waking life experiences, have been collected and used in their study. The target has been that of extract mentions of virtually any medical condition from text by applying deep learning algorithms. Their findings demonstrate how dreams reflected people's real-world experiences and that the health expressions that distinguished the two sets of data reflected differences in thought processes.

The idea of studying the dream reports along with social media posts and analyzing their impact on large-scale phenomena, such as the COVID-19 Pandemic, is extremely interesting. Moreover, this is the first study carrying out a systematic analysis of medical conditions featured in dream reports and compared them to those expressed in waking discussions, starting from the well-

known “continuity hypothesis of dreaming”. The latter refers to the relationship between health concerns discussed during waking life and the representation of such concerns in dreams, which is relevant to the study of people’s well-being.

As the same authors underline, one of the main limitations is in terms of biases linked with the data collection process, but they used a rank-based method to compare the health mentions in the two datasets, by ranking conditions by their frequency in each dataset separately.

In terms of methodology, they used MedDL for extracting mentions of medical conditions from text, and they trained the method on social media data, and as the same authors underline, training on dream reports could be beneficial for the analysis results.

Although the methodology is not novel in terms of NLP deep-learning tool used for extracting mentions of medical conditions, and the data collected and training process may produce biases in results, both the work and findings are novel.

The main aspect on which the authors could focus is on comparing the tool used with other methods to better validate the model. Moreover, a figure showing the pipeline of the methodology would help in the readability.

Finally, to strengthen the methodological approach and extending results, starting from the co-occurrence network of dream reports, which resembles the comorbidity networks (see the work: Moni, Mohammad Ali, and Pietro Liò. "Network-based analysis of comorbidities risk during an infection: SARS and HIV case studies." *BMC bioinformatics* 15.1 (2014): 1-23.), authors could explore some measures, such as the relative risk and ϕ -correlation. Respectively, these measures represent the relationship between the two conditions related to the datasets and the robustness of this association.

===PREPARING YOUR MANUSCRIPT===

If you have been asked to revise the written English in your submission as a condition of publication, you must do so, and you are expected to provide evidence that you have received language editing support. The journal would prefer that you use a professional language editing service and provide a certificate of editing, but a signed letter from a colleague who is a fluent speaker of English is acceptable. Note the journal has arranged a number of discounts for authors using professional language editing services (<https://royalsociety.org/journals/authors/benefits/language-editing/>).

===PREPARING YOUR REVISION IN SCHOLARONE===

Author's Response to Decision Letter for (RSOS-211080.R0)

See Appendix A.

Decision letter (RSOS-211080.R1)

Dear Dr Aiello,

It is a pleasure to accept your manuscript entitled "Epidemic Dreams: Dreaming about health during the COVID-19 pandemic" in its current form for publication in Royal Society Open Science. The comments of the reviewer(s) who reviewed your manuscript are included at the foot of this letter.

Please ensure that you send to the editorial office an editable version of your accepted manuscript, and individual files for each figure and table included in your manuscript. You can send these in a zip folder if more convenient. Failure to provide these files may delay the processing of your proof.

COVID-19 rapid publication process:

We are taking steps to expedite the publication of research relevant to the pandemic. If you wish, you can opt to have your paper published as soon as it is ready, rather than waiting for it to be published the scheduled Wednesday.

This means your paper will not be included in the weekly media round-up which the Society sends to journalists ahead of publication. However, it will still appear in the COVID-19 Publishing Collection which journalists will be directed to each week (<https://royalsocietypublishing.org/topic/special-collections/novel-coronavirus-outbreak>).

If you wish to have your paper considered for immediate publication, or to discuss further, please notify openscience_proofs@royalsociety.org and press@royalsociety.org when you respond to this email.

on behalf of Professor Yulan He (Associate Editor) and Marta Kwiatkowska (Subject Editor)
openscience@royalsociety.org

Appendix A

Response to the reviews of paper RSOS-211080

“Epidemic Dreams: Dreaming about health during the COVID-19 pandemic”

We would like to express our sincere thanks to the Associate Editor and the reviewers for their comments. We have worked to address all their concerns in the revised version of the manuscript. The main changes are highlighted with blue text. Below, we provide a detailed response to the reviewers' requests.

We hope they will find the new version of the paper improved.

Associate Editor's requests

While the reviewers found that the idea of studying the dream reports along with social media posts to conduct qualitative analysis of dreams during the pandemic is interesting, they raised a few major concerns that should be addressed in the revised manuscript. Examples include more accurate use of terminologies (such as REM sleep, continuity hypothesis), discussions of findings on dream activities in comparison to the existing descriptive data, more empirical studies to support some of the statements in the manuscript, additional measures to quantify the associations revealed by the co-occurrence network of dream reports, etc. Please refer to the reviews for the details of the major points raised

We have thoroughly revised the paper according to the Associate Editor's and the Reviewer's suggestions. In summary:

- We revised our terminology to define key concepts more accurately and unambiguously.
- We acknowledged prior survey-based work on pandemic dreams in the introduction, and discussed the similarities of those studies' results with our findings in the discussion.
- We rephrased the explanation of why dream reported contained metaphoric representations.
- We improved the visual summary of our analytics pipeline.
- We calculated two additional metrics to assess the robustness of the semantic associations between medical conditions in the network. We used those metrics to filter out spurious associations, and repeated the network analysis on the resulting network.

Requests from the Reviewer 1

The equivalence between REM sleep and dreaming is simply unacceptable. The authors have to realize that empirical research on dreams has definitely clarified that dreams are not exclusive of REM sleep and

that they can be collected practically in any stage of sleep. This means that all points in the manuscript in which they speak on REM sleep as a physiological scenario of dream production should be changed accordingly

The reviewer is right in noting that dreaming can occur in both REM and non-REM sleep cycles. We changed all the parts in the manuscript to remove or rephrase all the sentences that hinted at an exclusive association between REM sleep and dreams.

Since a large body of cross-sectional and longitudinal studies on the pandemic has been published using surveys, these findings on dream activity should be discussed compared to the current descriptive data. [...] The dramatic increase of nightmares during the pandemic is one of the most consistent findings of empirical studies and have a relevant clinical implication in relation to PTSD symptoms. The current qualitative findings on nightmares should also be discussed with reference to this evidence.

We thank the reviewer for providing references relevant to our work. We were aware of some of these papers—in the introduction of the original manuscript, we had already referenced “Pandemic nightmares” by Scarpelli et al. (reference 25)—but indeed we did miss some key contributions that were published later in 2021. To better acknowledge prior research that used surveys to quantify sleep and dreams during the pandemic, we expanded the list of references in the introduction with those suggested by the reviewer, and clarified how our contribution can enrich the perspective of survey-based studies. Unlike survey studies, the text of the dream reports was available to us, which allowed us to study mentions of medical conditions at a high level. We expanded the discussion to highlight similarities with the findings from previous literature. In particular, similar to our results, survey participants reported increased emotional load and higher frequency of nightmares and bizarre images compared to pre-lockdown. Participants also reported a substantial increase in anxiety, in agreement with the themes of panic, anxiety, phobia, and fear that we identified in our thematic analysis of network clusters. Pandemic-related dreams were also more frequent, according to survey respondents. In this work, we thoroughly analyzed the set of medical conditions in “pandemic dreams”.

Concerning the “continuity hypothesis”, it commonly refers to: (1) dreams are a continuation of our waking experiences; (2) neurobiological mechanisms underlying episodic memory in wakefulness are mostly similar to those of dream recall during sleep (e.g., Scarpelli et al. Investigation on Neurobiological Mechanisms of Dreaming in the New Decade. Brain Sci. 2021 Feb 11;11(2):220. doi: 10.3390/brainsci11020220). The article only mention the first meaning of this accepted meaning

In the revised version, we mention both meanings, and reference the suggested paper.

Please, substantiate the sentence in the discussion (page 10, rows 21.22): “The phrases more frequent in the dream reports about bizarre body disfunctions represent a metaphoric manner of thinking about COVID-19 with more activation in visual and emotional areas and less in verbal and logical ones.” The issue of greater activation in visual and emotional areas is still debated. Please, provide references. Still

more, I think that there is no empirical study showing that there is smaller activation "in verbal and logical ones."

The issue of higher/lower activation of emotional vs. logical areas is indeed debated. On the conservative side, we removed the sentence referring to brain activation altogether.

Requests from the Reviewer 2

The main aspect on which the authors could focus is on comparing the tool used with other methods to better validate the model.

We thank the reviewer for this comment. In the original version of the manuscript, we failed to provide a measure of quality of the output of our medical entity extractor. In previous work (Šćepanović et al. "Extracting medical entities from social media" ACM CHIL, 2020), we tested MedDL on standard benchmarks against the state-of-the-art methods for medical entity extraction from unstructured text. When applied to tweets, MedDL achieved an F1-score of 0.74; the baselines ranged from 0.34 to 0.63. When applied to Reddit posts, whose length and structure are more similar to dream reports compared to tweets, MedDL fared an F1-score of 0.71, while the competing approaches scored between 0.17 and 0.38. In short, MedDL outperformed by a large margin the state-of-the-art approaches developed until 2020 (the year MedDL was developed and published). We reported these statistics in the revised version of the paper.

A figure showing the pipeline of the methodology would help in the readability.

We completely revised Figure 1 to add detail to the representation of the pipeline. We hope that the reviewer will find this illustration improved.

To strengthen the methodological approach and extending results, starting from the co-occurrence network of dream reports, which resembles the comorbidity networks (see the work: Moni, Mohammad Ali, and Pietro Lió. "Network-based analysis of comorbidities risk during an infection: SARS and HIV case studies." BMC bioinformatics 15.1 (2014): 1-23.), authors could explore some measures, such as the relative risk and fi-correlation. Respectively, these measures represent the relationship between the two conditions related to the datasets and the robustness of this association.

We took the suggestion of the reviewer onboard and we calculated both the relative risk (RR_{ij}) and the Phi-correlation (ϕ_{ij}) between pairs of co-occurring conditions, and used them to filter out weak associations using the procedure from Ali and Lió, which we summarized in Section 3.2.3. The filtering step reduced the network from 1,419 nodes and 4,084 edges to 1,416 nodes and 3,759 edges. We repeated the analysis using this filtered network and updated the statistics in Section 4.2. The results varied only slightly, and the overall conclusions remained unchanged.

Some positive comments

The authors present a novel work on dream reports, highlighting their importance, if interpreted correctly, to better understand and get insights about people's health experiences in the real world, specifically in the case of extremely challenging and distressing events, such as the COVID-19 pandemic.

The idea of studying the dream reports along with social media posts and analyzing their impact on large-scale phenomena, such as the COVID-19 Pandemic, is extremely interesting.

this is the first study carrying out a systematic analysis of medical conditions featured in dream reports and compared them to those expressed in waking discussions.

We thank the reviewers for their positive comments.